# Epigenetic Regulation in Wilms Tumor

**DOI:** 10.3390/biomedicines13071678

**Published:** 2025-07-09

**Authors:** Annabelle Bolitho, Hongbing Liu

**Affiliations:** Department of Pediatrics, Tulane University School of Medicine, New Orleans, LA 70112, USA; abolitho@tulane.edu

**Keywords:** Wilms tumor, epigenetic regulatin, cancer intiation and progression, diagnostic and prognostic biomarkers, epigenetic therapies

## Abstract

Wilms tumor (nephroblastoma), the most common pediatric renal malignancy, has a complex genetic and epigenetic landscape. While mutations in genes like WT1, CTNNB1, and WTX have been well characterized, accumulating evidence suggests that epigenetic dysregulation plays a pivotal role in WT pathogenesis. This review examines the various epigenetic mechanisms implicated in WT, including DNA methylation, histone modifications, chromatin remodeling, and non-coding RNA-mediated regulation. We discuss how epigenetic mechanisms contribute to tumor initiation, progression, and heterogeneity and their implications for improved diagnosis and targeted therapy. We also highlight recent advances in epigenomic profiling, discuss the interplay between epigenetics and developmental gene expression programs, and evaluate potential therapeutic strategies targeting epigenetic regulators.

## 1. Introduction

Nephroblastoma, commonly known as Wilms tumor (WT), is the most common pediatric kidney cancer, typically affecting children at 3–5 years old [1,2]. Although there are excellent survival rates for WT, around 15% of patients do not respond to treatment, and this percentage increases as WT becomes more aggressive [1,3]. Current treatments, like chemotherapy, can cause long-term side effects, so it is critical to develop novel therapies to maintain high survival rates with fewer adverse effects. Alternative therapeutic approaches, such as surgery and radiation, have been employed for WT [4], particularly in cases of diffuse anaplasia, which is associated with a more severe prognosis and greater resistance to treatments [5]. Increasing evidence suggests that WT development is linked to aberrant nephrogenesis, which arises when embryonic kidney cells fail to mature into nephron progenitor cells around the ureteric bud [2,6,7,8]. The aberrant proliferation of these undifferentiated cells leads to tumor formation in the kidney cortex [2]. There have been substantial advances in our understanding of WT pathogenesis over recent decades; however, the mechanisms that disrupt differentiation and lead to tumor growth remain unclear. Although WT is associated with certain birth defects and gene mutations, there is no evidence supporting direct inheritance from parents. WT is genetically heterogeneous; mutations in Wilms tumor 1 (*WT1*), the Wilms tumor gene on the X chromosome (*WTX*; also known as *FAM123B* and *AMER1*), β-catenin (*CTNNB1*), and *TP53* have been identified in tumors [7,9,10]. While *WT1* at 11p13 (a specific location on the short arm (p arm) of chromosome 11, specifically band 13) was the first identified gene associated with the tumor, it is mutated in only a subset of cases (around 20%) [2,9,10]. Subsequent research identified other genes involved, including *WTX* at Xq11.1 (a specific band on the long arm (q arm) of the X chromosome) and CTNNB1, which collectively account for roughly one-third of WT when considered alongside *WT1* [10]. Somatic mutations at 11p15 have been discovered in 69% of tumors [11]. According to Fiala et al., patients predisposed to Beckwith–Wiedemann syndrome (BWS) on locus 11p15.5 (a specific region on the short arm of human chromosome 11, which is known to contain a cluster of imprinted genes) are at higher risk for WT; 28% of patients with a gain-of-function methylation at imprinting control 1 (IC1) develop WT [12]. In one sample of 21 WT patients, 33% were shown to have an epigenetic abnormality on locus 11p15.5 that could be detected in the blood [12]. This discovery indicates that epigenetic changes that lead to WT can be detected outside the tumor itself. Epigenetic alterations are central to oncogenesis in many pediatric cancers due to their low mutational burden [13]. WT is generally considered a non-familial disease, with epigenetic abnormalities significantly contributing to tumorigenesis [14]. WT is most often discovered in nephrogenic rests, which are consistently located around the renal medulla and periphery of the kidney, both of which arise during early renal development [6]. WT shares many features with the embryonic kidney, suggesting a close link between normal kidney development and the genesis of WT. The structure of the developing kidney is often maintained in WT, allowing us to study how WT interferes with normal kidney development as it progresses [6]. This review will summarize the significant contributions of epigenetic mechanisms to Wilms tumorigenesis compared to healthy kidney development, informing the development of novel and impactful therapies that alleviate adverse outcomes for affected children.

## 2. Epigenetic Alterations in Wilms Tumor

Epigenetics refers to changes in organisms driven by external factors—such as environmental influences and chemical exposures—that do not alter the organism’s DNA sequence. Both epigenetic and genetic changes play crucial roles in cancer development and progression. Epigenetic modifications, such as DNA methylation, histone modifications, and non-coding RNA, can alter gene expression and contribute to uncontrolled cell proliferation and other common characteristics of cancer, such as suppressed differentiation gene expression and immune evasion. These changes can be reversed, making epigenetics a promising avenue for cancer treatment. Epigenetic mechanisms are particularly relevant in childhood cancers due to their low rate of genetic mutations. WT is increasingly considered to be driven by epigenetic dysfunction [15]. Understanding the epigenetic basis of WT is critical in revealing the molecular mechanisms underlying pediatric kidney cancer and abnormal renal development. Although adult cancers are driven primarily by genetic mutations, WT often involves disruptions in developmental gene regulation through epigenetic alterations like loss of imprinting at 11p15 (IGF2/H19), DNA methylation changes, and mutations in chromatin remodeling genes (e.g., WT1, ARID1A, BCOR) [7]. These changes can lead to the persistence of undifferentiated nephron progenitor cells and dysregulated growth signals, contributing to tumorigenesis. Understanding these epigenetic mechanisms can improve risk stratification, identify biomarkers for early detection, and inform novel, targeted therapies that reverse epigenetic dysregulation to improve outcomes for children with WT.

## 3. DNA Methylation in Wilms Tumor

In DNA methylation, a major mechanism of epigenetic gene silencing, methyl groups are added to the fifth carbon of a cytosine base within a CpG dinucleotide (a pair of cytosine and guanine nucleotides connected by a phosphodiester bond). Methylation is carried out by enzymes called DNA methyltransferases, which transfer a methyl group onto cytosine bases. This modification reduces gene expression in the affected DNA strands, effectively silencing specific genes and regulating cellular function [16]. DNA methylation significantly contributes to cancer development and progression. Aberrant DNA methylation patterns, including both global hypomethylation and site-specific hypermethylation, are hallmarks of many tumors. These changes can lead to the silencing of tumor suppressor genes or the activation of oncogenes, contributing to uncontrolled cell growth and tumor formation [17]. In WT, altered methylation patterns can affect genes involved in tumor growth, development, and overall tumor behavior [18]. Increased DNA methylation is known to raise the risk of Wilms tumorigenesis through the 11p15.5 and 11p13 regions of the chromosome [12]. In particular, 11p15.5 harbors the WT2 locus, which includes imprinted genes like insulin-like growth factor 2 (IGF2) and H19 (a long non-coding RNA). Imprinting refers to the phenomenon where the expression of certain genes is determined by which parent the gene is inherited from. In WT, alterations in the imprinted genes at 11p15, like H19 and IGF2, can disrupt normal imprinting patterns and lead to the increased expression of IGF2, which can contribute to tumor growth [19]. H19 and IGF2 are transcriptionally active from the maternal and paternal chromosomes, respectively. A base methylation level of the H19/IGF2:IG-DMR genomic region allows H19 to be produced solely by the maternal allele and IGF2 by the paternal allele. When this region is hypermethylated along the maternal chromosome, IGF2 becomes transcriptionally active on the maternal allele and induces the inactivation of H19, a defect along the nephrogenic rests that also contributes to Wilms tumorigenesis [19]. The 11p13 region harbors the WT1 locus, and mutations in the WT1 gene are frequently observed in WT. Loss of heterozygosity (LOH) in this region can also disrupt WT1 function, contributing to Wilms tumorigenesis. WT can be classified into different subtypes based on DNA methylation patterns, which correlate with clinical outcomes. For example, one study identified four prognostic subtypes of WT, with differences in prognosis, age, sex, histological type, and tumor stage [20].

DNA methyltransferases (DNMTs) may drive unregulated cell proliferation in WT by inhibiting the expression of certain genes. With a more thorough understanding of the mechanisms behind methylation and its role in tumorigenesis, researchers can develop targeted approaches for potential therapies focused on reversing or modifying these epigenetic changes. Hypermethylation among certain genes, such as CCL2, CCL5, and CD4, has been linked to carcinogenesis, suggesting the potential for the discovery of DNA methyltransferase inhibitors [21]. These insights are crucial in understanding the role of methylation and identifying specific DNA modifications, allowing researchers to target genes and chromosomal regions of interest. By pinpointing these genetic loci, scientists can explore key developmental pathways to develop drugs that can demethylate hypermethylated regions. This approach may reduce unregulated cell proliferation, offering a promising therapeutic strategy for WT and other conditions. [18]. DNMT inhibition, by reactivating silenced genes, may lead to the expression of genes involved in tumor development, potentially helping to control tumor growth and reduce the inflammatory microenvironment in WT [18].

In summary, understanding the role of DNA methylation in WT development and progression is crucial in developing new diagnostic and therapeutic strategies.

## 4. RNA Methylation in Wilms Tumor

RNA methylation is a post-transcriptional modification in which methyl groups are added to RNA nucleotides, primarily at the N6 position of adenosine (m6A). Like DNA methylation, m6A is reversible and regulated by specific enzymes: writers (methyltransferases, such as METTL3, METTL14), erasers (demethylases, such as FTO, ALKBH5), and readers (YTHDFs and IGF2BPs) [22]. These modifications affect various aspects of RNA function, including splicing, translation, stability, and localization. Although other types of RNA methylation exist, m6A is the most prevalent and well studied. m6A plays a multifaceted role in cancer, influencing processes like cell proliferation, invasion, and metastasis. It affects both coding and non-coding RNAs, promoting or suppressing tumor growth depending on the context. m6A methylation can affect the tumor microenvironment and even influence cancer treatment resistance [23]. Abnormal m6A modifications in cancer can lead to changes in RNA stability, gene expression, protein translation, and RNA stability, ultimately contributing to cancer cell behavior [23,24].

Genetic variations in m6A modification genes like ALKBH5 have been investigated for their potential contribution to WT risk [25]. WT 1-associated protein (WTAP) is a component of the m6A writer complex, which is responsible for depositing m6A modifications on mRNA. WTAP specifically helps to recruit the m6A methyltransferase complex (METTL3 and METTL14) to target mRNAs [26]. WTAP may play a role in the m6A writer complex by regulating RNA metabolism and contributing to the development or progression of WT. Ongoing research aims to understand the specific mechanisms by which m6A modification, including that mediated by WTAP, contributes to WT development and to explore potential therapeutic targets based on m6A modification [27]. m6A may disrupt the finely tuned expression of genes critical for kidney development, contributing to the developmental arrest seen in WT. For example, m6A-modified transcripts involved in nephron progenitor maintenance (like SIX2-related pathways) may be improperly degraded or overexpressed [28]. IGF2, already central to WT via loss of imprinting at 11p15, is also regulated post-transcriptionally by m6A readers (IGF2BPs), suggesting the convergence of epigenetic and post-transcriptional control. Studies have revealed the potential prognostic value of m6A-related genes and their relationship with the immune microenvironment in WT [25]. Four m6A-related genes were successfully screened, including ADGRG2, CPD, CTHRC1, and LRTM2 [25]. Kaplan–Meier survival curves showed that the four genes were closely related to the prognosis of WT, which was also confirmed by receiver operating characteristic curves [25]. m6A machinery components are potential drug targets, and inhibitors targeting the METTL3 or IGF2BP proteins are under investigation in other cancers [29]. Targeting m6A dysregulation could offer a novel strategy for therapy-resistant WT.

Understanding the role of m6A RNA methylation in WT offers a novel perspective on post-transcriptional regulation in pediatric kidney cancer and may reveal new biomarkers and therapeutic targets for improved clinical outcomes. As our knowledge deepens, m6A-related pathways hold promise not only as biomarkers for diagnosis and prognosis but also as potential therapeutic targets in the pursuit of precision medicine for WT.

## 5. Histone Modifications in Wilms Tumor

Histone modifications are fundamental epigenetic mechanisms that alter the chromatin structure and regulate gene expression. Histones are the main proteins around which DNA is wrapped in the nucleus, forming nucleosomes. Histone modifications involve covalent alterations to histone proteins, primarily acetylation, methylation, and phosphorylation, all of which can regulate transcription by altering the chromatin structure, directly affecting how tightly DNA is packed, and by recruiting other proteins that influence gene expression. Specific enzymes, like histone acetyltransferases (HATs), histone deacetylases (HDACs), histone methyltransferases, and histone phosphatases, are responsible for adding or removing these modifications [30]. Histone modifications play a significant role in cancer development and progression [31]. Dysregulation of these modifications can lead to aberrant gene expression, disrupting cell processes and contributing to tumor formation, invasion, and metastasis [32].

Histone acetylation and deacetylation, the key epigenetic processes, play a crucial role in regulating gene expression and are implicated in various diseases, including cancer. HATs add acetyl groups to histones, leading to increased gene expression, while histone HDACs remove acetyl groups, resulting in gene silencing. In WT, an imbalance in histone acetylation and deacetylation can lead to the aberrant expression of genes involved in tumor growth and progression. For example, the downregulation of tumor suppressor genes due to increased deacetylation can contribute to tumor development. MYCN, a key oncogene, is linked to histone acetylation, in which acetyl groups are added to histone tails, altering the chromatin structure and affecting gene expression [33]. MYCN alters histone acetylation patterns, influencing gene expression and promoting tumor development. It interacts with HATs/HDACs to modulate oncogenic gene expression, and MYCN amplification is a key characteristic of aggressive WT, particularly the anaplastic subtype [34]. *MYCN* amplification dysregulates cell cycle progression, leading to relapse, poor survival, and treatment resistance [35]. Anaplasia in WT is defined as threefold nuclear enlargement, hyperchromasia, and abnormal mitotic figures [30]. HDACs can also deacetylate non-histone proteins, influencing cell proliferation, apoptosis, and differentiation. Studies have revealed the critical role of HDACs, especially HDAC1 and HDAC2, in kidney development [36,37,38,39]. Some studies suggest that HDACs (particularly HDAC1-2 and HDAC4-5) are overexpressed in WT, contributing to tumorigenesis by silencing tumor suppressor genes [40,41,42]. HDAC inhibitors have shown promise in treating WT, especially in high-risk cases like blastemal-predominant WT [42]. These inhibitors, like panobinostat and romidepsin, are effective against a variety of WT types. Research has shown that HDAC inhibitors can target specific oncogenic pathways in WT, potentially improving treatment outcomes [42,43]. Wilms tumor 1 (WT1) is a gene that encodes a protein involved in regulating cell growth and differentiation. WT1 interacts with HATs (e.g., p300/CBP) and HDACs to regulate gene expression [44,45,46]. Mutations or loss of WT1 were found to disrupt the acetylation balance, leading to aberrant proliferation [47]. Loss of imprinting (LOI) of IGF2 is common in WT. One study also showed that HDAC inhibition can restore normal imprinting, suggesting a role for acetylation in IGF2 regulation [48]. In short, histone acetylation/deacetylation imbalances contribute to WT pathogenesis, particularly through WT1 dysfunction, IGF2 dysregulation, and HDAC overexpression. Targeting these epigenetic mechanisms could offer new treatment strategies.

Histone methylation refers to the addition of methyl groups (–CH_3_) to specific amino acids on histone proteins (like lysine or arginine residues on histones H3 and H4). These modifications alter the chromatin structure and regulate gene expression, either activating or repressing transcription depending on the site and degree of methylation. Histone methylation is catalyzed by histone methyltransferases (HMTs) and removed by histone demethylases (KDMs). In cancer, histone methylation is often dysregulated, leading to the abnormal expression of oncogenes or tumor suppressor genes. Histone methylation plays a crucial role in WT development and progression, influencing gene expression and potentially contributing to tumor growth and metastasis. In WT, the dysregulation of HMTs and KDMs contributes to altered gene expression and tumor development. Enhancer of Zeste Homolog 2 (EZH2) is the catalytic subunit of Polycomb Repressive Complex 2 (PRC2), which tri-methylates H3K27 (H3K27me3) to repress gene transcription. EZH2 plays a crucial role in kidney development and disease [49] and is often overexpressed in WT, leading to excessive H3K27me3 at key differentiation gene promoters, the suppression of genes critical for kidney cell differentiation, and the maintenance of progenitor-like, undifferentiated states in nephron progenitor cells (especially Six2+ cells) [49]. As a result, kidney development is blocked at an immature stage, and proliferation continues, promoting tumor formation. Studies show that high EZH2 expression correlates with poor differentiation in WT, and some aggressive subtypes of WT (e.g., anaplastic WT) show especially high EZH2 activity. EZH2 is increasingly being studied as a biomarker (Table 1). Drugs targeting histone methylation, like EZH2 inhibitors (e.g., tazemetostat), are now being developed and approved for certain cancers (e.g., epithelioid sarcoma, follicular lymphoma) [50].

H3K4me3 is a histone modification in which lysine 4 on histone H3 is methylated three times. It is generally considered a marker of active chromatin and is often found at promoter regions and transcription start sites. In the context of WT, the regulation of H3K4me3, particularly by enzymes like MLL (also known as KMT2A), plays a role in tumor development. Studies have shown increased levels of H3K4me3 in WT [6,52]. The elevated H3K4me3 in WT may contribute to the upregulation of genes involved in cell proliferation, differentiation, and growth, contributing to tumor growth and progression. H3K4me3 marks are enriched at promoters of genes involved in cell proliferation, stemness, and developmental pathways (e.g., *MYCN*, *LIN28B*) [53]. The overexpression of these genes due to aberrant H3K4me3 can drive WT growth [6,54]. Loss of H3K4me3 at tumor suppressor genes (e.g., *TP53*) may contribute to their silencing [55]. Similarly, WT1, frequently mutated in WT, interacts with the histone methylation machinery and influences H3K4me3; loss of WT1 function reduces this activation landscape at key differentiation genes, blocking nephron maturation and promoting tumor growth [56]. Mutations in H3K4 methyltransferases (e.g., MLL1-4, SETD1A/B) or demethylases (e.g., KDM5 family) can disrupt normal kidney development [57,58]. WT often exhibits abnormal retention of bivalent marks, silencing genes crucial for renal differentiation and causing aberrant transcriptional programs [57,58]. Bivalent domains are genomic regions marked by both H3K4me3 (activation signal) and H3K27me3 (repression signal). These are common in embryonic stem cells (ESCs) and progenitor cells, keeping genes poised for activation or silenced during differentiation [6]. The interaction between H3K27me3 and H3K4me3 is a master regulator of WT biology, influencing stemness, differentiation, and aggressiveness. KDM6A, a demethylase that removes H3K27me3 and promotes gene activation, may be underexpressed or functionally impaired, leading to the persistent repression of critical developmental regulators. These alterations highlight a complex mechanism of histone methylation dynamics that drives WT pathogenesis and offers promising targets for epigenetic therapies to restore proper chromatin states.

Histone modification dysregulation is increasingly recognized as a critical contributor to WT development and progression. Aberrant patterns of histone acetylation, methylation, and other post-translational modifications disrupt normal gene expression programs that are essential for renal differentiation, leading to uncontrolled proliferation and tumorigenesis. Among these modifications, the activity of HDACs has emerged as particularly important, as their aberrant function can silence tumor suppressor genes and maintain the undifferentiated state of tumor cells. Targeting histone-modifying enzymes, especially through HDAC inhibitors, offers a promising therapeutic strategy for the restoration of normal epigenetic control and inhibition of tumor progression. Continued research into the specific histone modification patterns and their regulatory enzymes in Wilms tumor will be essential in developing more effective and targeted epigenetic therapies.

## 6. Non-Coding RNA

Non-coding RNA (ncRNA) refers to RNA molecules that are not translated into proteins, unlike messenger RNA (mRNA), which carries code for protein synthesis. ncRNAs regulate gene expression at various levels, including the chromatin structure, transcription, and post-transcriptional processes. They are dysregulated in various cancers and can act as oncogenes or tumor suppressors. These ncRNAs, including microRNAs (miRNAs), long non-coding RNAs (lncRNAs), and circular RNAs (circRNAs), influence multiple cancer hallmarks, like proliferation, apoptosis, invasion, and metastasis [59].

miRNAs are ~22-nucleotide RNAs that suppress gene expression by targeting messenger RNAs (mRNAs) for degradation or translational inhibition. miRNAs are crucial in the development of WT by regulating the expression of specific genes [60,61,62]. There is an increasing amount of research that connects the dysregulation of miRNAs to the development of various renal diseases. miRNAs could contribute to the development and progression of WT by controlling cell proliferation, migration, and apoptosis [60,61,62]. The miR-17-92 cluster, a group of microRNAs, is often overexpressed in WT and is implicated in promoting tumorigenesis. It has been shown to evade apoptosis, disrupt senescence, and enhance oncogenic transformation. Specifically, miR-17 and miR-20a within the cluster can inhibit senescence by targeting p21WAF1. The cluster can also suppress apoptosis by targeting PTEN through miR-19 components [63]. A recent study revealed that the expression of miR-204 in WT tissue was significantly lower than that in adjacent normal tissue [64], but tumor tissue showed higher miR-485-5p expression than the adjacent normal tissue [64]. miR-204 could be a novel biomarker for anaplastic tumors and may be a useful target for the development of therapeutic interventions [64]. The let-7 family of microRNAs plays an important tumor-suppressive role in many cancers, including WT [65]. Its dysregulation is a key feature of WT pathogenesis, often in connection with LIN28 overexpression. Mouse models overexpressing LIN28 in renal progenitors develop WT-like lesions, and these are characterized by low let-7 expression [65].

Alterations in miRNA biogenesis are closely linked to cancer development and progression, as disruptions in this pathway can significantly impact gene expression and cellular signaling networks that are critical for tumorigenesis. Such dysregulation may result from genetic mutations in core biogenesis genes, epigenetic modifications, or the abnormal activity of proteins involved in miRNA processing and maturation [51]. In WT, recurrent mutations in key miRNA processing components, most notably DROSHA and DICER1, have been identified. These mutations impair the maturation of specific miRNAs, including tumor-suppressive members of the *let-7* family, which regulate important oncogenic targets such as *MYCN* and *LIN28* [52,53]. Consequently, defects in miRNA biogenesis not only contribute to disrupted gene regulation but may also define a distinct molecular subtype of WT with unique pathogenic features. Recent studies further demonstrate that DROSHA mutations in WT, as well as DROSHA silencing in vitro, are associated with a mesenchymal-like state and the dysregulation of redox metabolism, suggesting broader effects on tumor biology beyond miRNA processing [61].

While most studies on WT have focused on profiling miRNA signatures within tumor biopsies, there is increasing evidence that circulating miRNAs isolated from body fluids—such as serum, plasma, or whole blood—can serve as informative, non-invasive biomarkers for WT diagnosis and prognosis [66,67,68]. Over the past decade, the development of liquid biopsy approaches has gained significant attention, and the inherent stability of miRNAs in biofluids further supports their potential as reliable clinical biomarkers [69]. A qualitative analysis of 280 samples identified a core set of circulating miRNAs associated with WT development and progression, highlighting their strong diagnostic value [66]. Notably, recent serum miRNAome profiling in pediatric Egyptian patients with WT revealed that miR-180-3p may serve as a promising blood-based biomarker for distinguishing between WT histopathological subtypes [67].

Long non-coding RNAs (lncRNAs) are transcripts with over 200 nucleotides that regulate gene expression through chromatin remodeling, transcriptional control, and miRNA sponging. Several lncRNAs have been implicated in WT development, including LINC00473, SOX21-AS1, LINC00667, SNHG6, HOXA11-AS, MYLK-AS1, and XIST [70]. These lncRNAs have been shown to play different roles in WT development, including promoting cell proliferation, inhibiting apoptosis, and regulating cell cycle progression [71]. TUG1 is a long non-coding RNA (lncRNA) that has been implicated in various cancers. Studies suggest that TUG1 expression may be altered in WT, potentially influencing tumor cell growth, migration, and invasion [72]. While there is no direct TUG1-based treatment for WT, research indicates that it may play a role in tumor development and progression [73]. In WT, the lncRNA MALAT1 is found to be downregulated in tumor tissue compared to normal tissue [74]. This downregulation suggests that MALAT1 may act as a tumor suppressor or a prognostic biomarker for Wilms tumor. Additionally, MALAT1 expression can be detected in exosomes, which are extracellular vesicles that carry cargo, including nucleic acids and proteins for cellular communication [74].

Circular RNAs (circRNAs) are covalently closed-loop RNAs that act as miRNA sponges, regulate transcription, or interact with RNA-binding proteins. circCDYL is considered a tumor suppressor in WT, as its overexpression suppresses cell proliferation, migration, and invasion both in vitro and in vivo [75]. Emerging evidence suggests that circRNAs play important roles in pediatric cancers, including WT, where they modulate oncogenic and tumor-suppressive pathways. circCDYL acts by targeting miR-145-5p, a microRNA that regulates the expression of genes involved in tumor growth and metastasis [75]. Cao et al. (2021) found that circ0093740 was upregulated in WT cells and tissue [76]. Suppressing circ0093740 inhibits cell proliferation and migration [76]. Mechanistically, circ0093740 promotes tumor growth and metastasis by sponging miR-136/145 and upregulating DNMT3A, a DNA methyltransferase [76]. A study by Zhou et al. suggests that circRNAs are involved in the ceRNA (competing endogenous RNA) network in WT [75]. This network involves circRNAs, microRNAs, and mRNAs, where circRNAs act as miRNA sponges, regulating the expression of their target mRNAs [77]. Although circRNA research in WT is in its early stages, available evidence highlights their regulatory importance in tumor biology. Further experimental validation and clinical correlation studies are needed to fully elucidate their potential as biomarkers or therapeutic targets.

In short, non-coding RNAs have emerged as critical regulators of gene expression in WT, influencing key pathways involved in tumor initiation, progression, and differentiation. The dysregulation of non-coding RNAs in WT not only reflects underlying molecular alterations but also offers promising avenues for novel biomarkers and therapeutic targets. Continued investigation into the functional roles and regulatory networks of non-coding RNAs may provide valuable insights into WT biology and support the development of RNA-based diagnostic and treatment strategies.

As summarized in Figure 1, WT displays a complex epigenetic landscape that extends beyond the well-characterized loss of imprinting (LOI) at the *IGF2* locus. This includes aberrant DNA/RNA methylation, histone modifications, and the dysregulation of non-coding RNAs. These alterations frequently localize to specific genomic regions, such as chromosome 11p15, and collectively contribute to the initiation, progression, and heterogeneity of the tumor. Accumulating evidence has underscored the critical role of epigenetic regulation in the development and progression of WT. These epigenetic changes can disrupt key developmental pathways, reprogram cellular identity, and influence tumor behavior, often in a subtype-specific manner.

## 7. Therapeutic Implications of Epigenetic Regulation in WT

WT is one of the most curable pediatric malignancies, with overall survival rates exceeding 90% for patients with favorable histology [78]. Current treatments are risk-stratified and multimodal, combining surgery, chemotherapy, and radiotherapy where indicated. Surgical excision is the foundation of WT management. Most children undergo unilateral nephrectomy, although nephron-sparing surgery is increasingly used in bilateral or predisposed cases [79]. Chemotherapy regimens, typically involving vincristine, dactinomycin, and doxorubicin, are administered based on the tumor stage and histology using protocols informed by Children’s Oncology Group (COG) and International Society of Pediatric Oncology (SIOP) trials [78]. Radiation is used selectively for stage III/IV tumors, particularly those with anaplastic histology or pulmonary metastases. Radiation therapy improves local control but must be carefully balanced against long-term toxicity [1]. While WT has relatively few recurrent mutations, alterations in pathways like IGF2, WNT, and mTOR offer opportunities for molecular intervention. Despite promising evidence, these agents are not yet standard due to mixed efficacy and concerns over pediatric toxicity, and outcomes for patients with diffuse anaplasia, recurrence, or chemoresistance remain suboptimal, underscoring the need for more targeted and less toxic therapies.

Increasing evidence highlights the critical role of epigenetic mechanisms in the initiation and progression of WT. These epigenetic changes are reversible, making them attractive targets for therapeutic intervention. The genetic mutations of WT are diverse, and the mutated genes are often epigenetic regulators. Epigenetic dysregulation plays a pivotal role in WT genesis and provides a rationale for targets to treat WT. DNMT inhibitors, such as azacytidine (Vidaza) and decitabine (Dacogen), can disrupt DNA methylation patterns by inhibiting DNMT activity [80]. Some studies have specifically investigated the use of DNMT inhibitors in combination with other treatments for solid tumors, showing promising results in preclinical models and potentially in clinical trials [54,81]. While standard treatment involves surgery and chemotherapy, DNMT inhibitors could offer a new approach to address the genetic and epigenetic alterations that contribute to WT development and progression [18]. HDACs play a significant role in WT development and progression. The overactivation of HDACs, particularly HDAC1 and HDAC2, has been linked to the unrestrained proliferation of progenitor cells, a key factor in WT formation [41]. While specific, large-scale clinical trials dedicated solely to HDAC inhibitors for WT may be limited, research indicates that HDAC inhibitors show promise as a therapeutic approach for WT, particularly in combination with other targeted therapies like EZH2 inhibitors [41], and pre-clinical studies have demonstrated their efficacy in inhibiting tumor growth and inducing apoptosis in WT cells and models [42]. The less differentiated blastemal type of WT often relapses. Ma et al. generated a cell model of WiT49-PRCsP that recapitulated blastemal WT, leading to the discovery of therapeutic vulnerability for high-risk WT [42]. Drug screening identified the marked hypersensitivity of both epithelial- and blastemal-predominant WT patient-derived xenografts (PDXs) to HDAC inhibitors [42]. Among the compounds tested, panobinostat and romidepsin demonstrated the most potent antitumor effects across both histologic subtypes. Panobinostat, a pan-HDAC inhibitor, exhibited broad activity but was associated with increased toxicity, likely due to its inhibition of multiple HDAC isoforms. In contrast, romidepsin, which selectively targets HDAC1 and HDAC2, showed a more favorable therapeutic profile [42]. Several HDAC inhibitors are FDA-approved for cancer treatment and may offer a promising approach for the treatment of WT, especially high-risk, metastatic WT. The role of EZH2 in WT has led to research targeting this protein as a potential therapeutic approach. The EZH2 inhibitor tazemetostat has been FDA-approved for the treatment of sarcoma, and trials targeting pediatric solid tumors, including WT, are underway. Aberrant patterns of histone modifications identified in WT suggest the potential of epigenetic enzymes as therapeutic targets (Table 2) [42,82]. Epigenetic therapies may work synergistically (e.g., EZH2 + HDAC or DNMT + HDAC inhibitors) to restore differentiation, inhibit proliferation, and overcome drug resistance.

Epigenetic dysregulation plays a central role in the pathogenesis and heterogeneity of WT, particularly through the aberrant activity of chromatin-modifying enzymes such as HDACs and EZH2. These regulators contribute to the maintenance of an undifferentiated, progenitor-like tumor state by repressing genes involved in differentiation and tumor suppression. Preclinical studies demonstrating the sensitivity of WT models—especially blastemal-predominant subtypes—to HDAC and EZH2 inhibitors underscore the therapeutic potential of targeting these epigenetic mechanisms. Selective HDAC1/2 inhibitors like romidepsin and EZH2 inhibitors such as tazemetostat offer promising avenues for intervention with potentially fewer off-target effects. Combination strategies that concurrently inhibit multiple epigenetic regulators may further enhance tumor differentiation and reduce resistance. As our understanding of the epigenomic landscape in WT continues to evolve, integrating epigenetic therapies into precision treatment approaches holds significant promise in improving outcomes in high-risk and refractory disease.

## 8. Conclusions and Perspectives

Epigenetic regulation is a fundamental driver of WT biology. Dysregulation of DNA methylation, histone modifications, and non-coding RNAs all contribute to the disruption of normal nephrogenesis. Key epigenetic modifiers, including HDACs and EZH2, are aberrantly expressed in WT and contribute to the maintenance of a progenitor-like, undifferentiated state, particularly in the aggressive blastemal subtype. Recent advances in epigenetic profiling have identified potential biomarkers and therapeutic targets, offering new avenues for precision medicine in WT treatment. Accumulating evidence from preclinical studies suggests that targeting these epigenetic mechanisms can suppress tumor growth, promote differentiation, and enhance the therapeutic sensitivity. Selective inhibitors of HDAC1/2 and EZH2 are particularly promising and may offer more precise therapeutic options with reduced toxicity compared to broader epigenetic drugs.

The complex relationships between genetic and epigenetic alterations and their dynamic changes during tumor progression remain poorly understood. Future research should focus on integrating multi-omics approaches to reveal the epigenetic landscape of WT, allowing the development of targeted therapies that restore normal epigenetic control. Exploring the role of the tumor microenvironment and epigenetic plasticity could further improve clinical outcomes. Looking forward, the integration of epigenetic profiling into clinical diagnostics could refine risk stratification and identify patients most likely to benefit from targeted epigenetic therapies. Combination strategies that incorporate epigenetic modulators with conventional chemotherapy or emerging immunotherapies hold the potential to overcome resistance and improve long-term outcomes. Understanding how developmental epigenetic programs are subverted in WT may reveal novel targets and biomarkers. A deeper investigation into the tumor-specific epigenetic landscape will pave the way for novel diagnostic and prognostic biomarkers, ultimately advancing precision medicine in pediatric renal oncology.

## Figures and Tables

**Figure 1 biomedicines-13-01678-f001:**
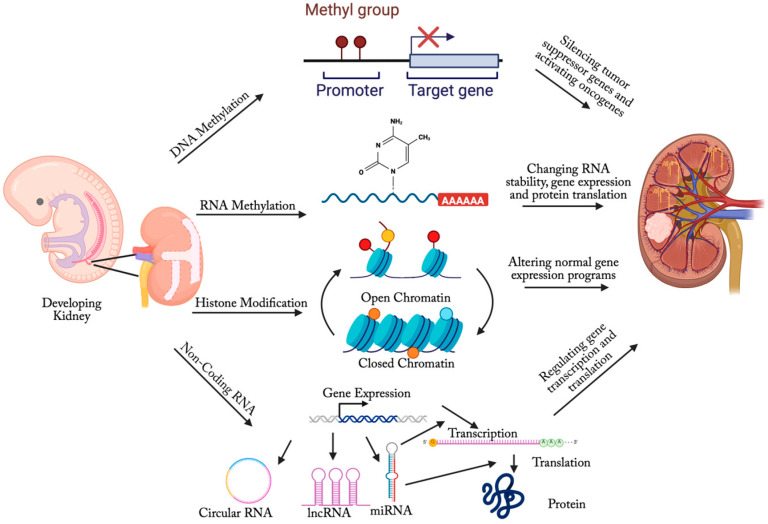
Aberrant epigenetic alterations in Wilms tumor development.

**Table 1 biomedicines-13-01678-t001:** EZH2 serves as both a biomarker for aggressive disease and a potential therapeutic target in WT.

Role	Details	Reference(s)
Diagnostic Biomarker	EZH2 overexpression is common in WT compared to normal kidney tissue. It can help distinguish tumor tissue from normal or benign kidney lesions in some cases.	[41]
Prognostic Biomarker	High levels of EZH2 correlate with poor prognosis in many cancers. In WT, higher EZH2 expression has been associated with worse differentiation and potentially more aggressive tumor behavior, but this link needs more clinical validation.	[51]
Predictive Biomarker	EZH2 expression could predict the response to EZH2 inhibitors in the future, although this is still at an experimental stage for WT.	[1,51]
Therapeutic Target	EZH2 is a marker and an active target: inhibiting it might reverse the block in differentiation that drives WT growth.	[41]

**Table 2 biomedicines-13-01678-t002:** Summary of histone modification inhibitor potential for the treatment of Wilms tumor.

Name	Target(s)	Mechanism	Potential Relevance in Wilms Tumor	Development Status
Panobinostat	HDAC (Class I, II, IV)	Broad-spectrum HDAC inhibitor; induces apoptosis, cell cycle arrest	May target aberrant epigenetics in Wilms tumor; preclinical studies needed	FDA-approved (myeloma)
Vorinostat (SAHA)	HDAC (Class I, II)	Promotes histone acetylation, reactivates tumor suppressor genes	Potential for differentiation therapy in Wilms tumor	FDA-approved (myeloma)
Romidepsin	HDAC (Class I)	Selective HDAC1/2 inhibition; disrupts oncogenic pathways	Possible synergy with chemotherapy in pediatric tumors	FDA-approved (myeloma)
Tazemetostat (EPZ-6438)	EZH2 (H3K27me3 methyltransferase)	Blocks PRC2-mediated silencing of tumor suppressors	EZH2 overexpression linked to poor prognosis; may inhibit Wilms tumor progression	FDA-approved (epithelioid sarcoma)
GSK126	EZH2 inhibitor	Reduces H3K27me3 marks, reactivates silenced genes	Preclinical potential for Wilms tumors with EZH2 dysregulation	Investigational
JQ1	BET (BRD4) inhibitor	Targets bromodomains; suppresses MYC and other oncogenes	MYC is implicated in Wilms tumor; may disrupt oncogenic transcription	Preclinical
CPI-203	BET inhibitor	Downregulates pro-proliferative genes	Potential for high-risk Wilms tumor subtypes	Investigational
Valproic Acid	HDAC (Class I, IIa)	Weak HDAC inhibitor; induces differentiation	Possible adjunct therapy due to low toxicity in pediatrics	FDA-approved (epilepsy)

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
