# Peer review of "Epigenetic Regulation in Wilms Tumor"

_biomedicines, 2025, doi:10.3390/biomedicines13071678_

Round 1

Reviewer 1 Report

Comments and Suggestions for Authors

Thank you for submitting your review of Wilm's Tumor.

As a reviewer I appreciate the authors including non-coding RNA as an epigenetic explanation for the genetic regulation that underlies Wilm's tumor development. 

In this section, please discuss the mutations in the miRNA biogenesis proteins such as Dicer and Drosha that have been reported to be associated with Wilm's tumor more that two decades ago. 

A second comment is to provide additional references. 60 something references is not sufficient for an in-depth review of Wilm's tumor.

Do the authors have any data on survival rate, drug resistance, and other clinical measures in patients on standard therapy +/- HDAC inhibitors?

I think another figure highlighting some of the mechanisms mentioned in the text that regulates the cellular pathways in the kidney that causes Wilm's tumor would be crucial to strengthening the quality of the manuscript.

Also there is no need to bold certain text words and not others. 

Comments on the Quality of English Language

n/a

Author Response

  1. In this section, please discuss the mutations in the miRNA biogenesis proteins such as Dicer and Drosha that have been reported to be associated with Wilm's tumor more than two decades ago. 

Thanks. The following discussion was added to the manuscript. “In WT, recurrent mutations in key miRNA processing components, most notably DROSHA and DICER1, have been identified. These mutations impair the maturation of specific miRNAs, including tumor-suppressive members of the let-7 family, which regulate important oncogenic targets such as MYCN and LIN28 [52–53]. Consequently, defects in miRNA biogenesis not only contribute to disrupted gene regulation but may also define a distinct molecular subtype of WT with unique pathogenic features. Recent studies further demonstrate that DROSHA mutations in WT, as well as DROSHA silencing in vitro, are associated with a mesenchymal-like state and dysregulation of redox metabolism, suggesting broader effects on tumor biology beyond miRNA processing [61].”

  1. A second comment is to provide additional references. 60 something references is not sufficient for an in-depth review of Wilm's tumor

Thanks. Now, 84 references have been cited for the review.

  1. Do the authors have any data on survival rate, drug resistance, and other clinical measures in patients on standard therapy +/- HDAC inhibitors?

Thanks. The following sentences were added to the manuscript. “While specific, large-scale clinical trials dedicated solely to HDAC inhibitors for WT may be limited, research indicates that HDAC inhibitors show promise as a therapeutic approach for WT, particularly in combination with other targeted therapies like EZH2 inhibitors [43], and pre-clinical studies have demonstrated their efficacy in inhibiting tumor growth and inducing apoptosis in WT cells and models [44].”

  1. I think another figure highlighting some of the mechanisms mentioned in the text that regulates the cellular pathways in the kidney that causes Wilm's tumor would be crucial to strengthening the quality of the manuscript.

Thanks. A figure (Figure 1) was included to highlight some of the mechanisms mentioned in the text.

  1. Also there is no need to bold certain text words and not others. 

Thanks. Corrected.

Reviewer 2 Report

Comments and Suggestions for Authors
    • The text reads quite dense and technical, which is fine for a scientific audience but might benefit from clearer transitions between topics.
    • Some sentences are long and packed with information; breaking them into shorter, clearer sentences would improve readability.
  1. Repetition and clarity:
    • The role of HDACs and EZH2 is discussed multiple times in very similar terms (e.g., their overexpression and role in progenitor-like states). Consider condensing or streamlining these points to avoid redundancy.
    • The phrase “WTigenesis” appears once and may be a typo or a term that needs clarification (is it a shorthand for “Wilms Tumorigenesis”?). Clarify or correct this.

Author Response

  1. The text reads quite dense and technical, which is fine for a scientific audience but might benefit from clearer transitions between topics

Thanks. Revised.

  1. Some sentences are long and packed with information; breaking them into shorter, clearer sentences would improve readability

Thanks. Revised.

  1. The role of HDACs and EZH2 is discussed multiple times in very similar terms (e.g., their overexpression and role in progenitor-like states). Consider condensing or streamlining these points to avoid redundancy.

Thanks. Revised.

  1. The phrase “WTigenesis” appears once and may be a typo or a term that needs clarification (is it a shorthand for “Wilms Tumorigenesis”?). Clarify or correct this.

Thanks. Corrected.